# Biases in neural population codes with a few active neurons

**Sander W. Keemink** [ID][1]**, Mark C.W. van Rossum** [ID][2,3]*

**1** Department of Machine Learning and Neural Computing, Donders Institute for Brain, Cognition and Behaviour, Radboud University, Nijmegen, The Netherlands, **2** School of Psychology, University of Nottingham, Nottingham, United Kingdom, **3** School of Mathematical Sciences, University of Nottingham, Nottingham, United Kingdom

* mark.vanrossum@nottingham.ac.uk

**Data availability statement:** Code is available at https://github.com/vanrossumlab/bias24.

**Funding:** The author(s) received no specific funding for this work.

## Abstract

Throughout the brain information is coded in the activity of multiple neurons at once, so called population codes. Population codes are a robust and accurate way of coding information. One can evaluate the quality of population coding by trying to read out the code with a decoder, and estimate the encoded stimulus. In particular when neurons are noisy, coding accuracy has extensively been evaluated in terms of the trial-to-trial variation in the estimate. While most decoders yield unbiased estimators if many neurons are activated, when only a few neurons are active, biases readily emerge. That is, even after averaging, a systematic difference between the true stimulus and its estimate remains. We characterize the shape of this bias for different encoding models (rectified cosine tuning and von Mises functions) and show that it can be both attractive or repulsive for different stimulus values. Biases appear for maximum likelihood and Bayesian decoders. The biases have a non-trivial dependence on noise. We also introduce a technique to estimate the bias and variance of Bayesian least square decoders. The work is of interest to those studying neural populations with a few active neurons.

## Author summary

The way information is represented in neurons is a fundamental property of computation in neural systems. In most brain regions, a single stimulus leads to the activity of multiple neurons. Such a so called population code combines high representational capacity with robustness against neural death and noise. Numerous studies have studied decoders of noisy population activity that minimize the trial-to-trial variance in the estimate, thereby revealing fundamental limits to the code. However, decoding can also be biased, that is, even in the limit of an infinite number of observations, a difference between the estimate and the actual value of a stimulus remains. Biases have been occasionally studied and appeared to emerge only in special situations (sub-optimal decoders, the presence of multiple stimuli, or non-uniform stimulus priors). Here we show that biases already emerge naturally when only a small population of neurons is active. The bias emerges for all common decoding methods. The biases have a complex

**Competing interests:** The authors have declared that no competing interests exist.

dependence on neural tuning curves and noise, but we develop an effective approximation technique for the Bayesian estimator decoder. The work is of importance for studying how neuron populations with few active neurons encode information.

## Introduction

In many brain areas neurons code information in the form of a population code. That is, a stimulus drives the activity of multiple neurons, thereby yielding codes that are robust to neural noise and neural death, while still having a high capacity [1]. As population codes are at the core of neural information processing, over the years many properties of population codes have been studied. A central theme of many of these studies has been the quality of the code and the decoder. Historically, this quality has been mainly quantified by the trial-to-trial variance in the decoded variable (which is similar to representational capacity). Using this metric, numerous studies have addressed questions, such as, how accurate are population codes in the presence of noise [2], how do correlations in the noise degrade information [3,4], and how can population codes be optimally transmitted [5].

In addition to trial-to-trial variance, a decoder can also exhibit a bias. That is, even after averaging over many trials a systematic difference between the true value of the encoded stimulus and its estimate remains. Given an estimate $\hat{\theta}$ and a true stimulus value $\Theta$, one defines bias as

$$b(\Theta) = \langle \hat{\theta} \rangle - \Theta$$

where the angular brackets denote averaging over trials.

In the limit of many active neurons with low noise and with perfect statistical knowledge of the encoding model, commonly used decoders have no bias [6]. But biases can emerge when these assumptions are not met. For instance, biases emerge when the encoder adapts but the decoder does not, so that the decoder model does not match the encoder [7]. Biases can also emerge when multiple stimuli are coded simultaneously in a neural population [8,9]. Biases have also been studied in Bayesian perception when the stimulus priors are non-uniform [10].

It might thus appear that biases only occur in special cases. However, as we will see, biases arise already in much simpler scenarios, namely in neural populations with sparse activation. We study this scenario by using low number of neurons and/or narrow tuning curves. We characterize the biases that arise for several different decoders and noise levels, and consider the theoretical and experimental implications.

## Results

### Encoding model

While many decoders are unbiased under assumptions of many neurons [6], biases emerge when the stimulus activates only a limited number of neurons. This might be the case because tuning curves are narrow (see below), or because the population is small, such as happens in insects. We start with analyzing the latter case.

A common example of a low-dimensional population code is the cricket wind sensor system [11–13]. Here an angular stimulus $\Theta$, the wind direction, is encoded by just four neurons, $k = 1 \ldots N = 4$. We assume that the neurons have preferred stimuli $\phi_k$ that are spaced 90 degrees apart ($\phi_k = k\pi/2$) and have a rectified cosine tuning

$$f_{k=1\ldots4}(\Theta) = \frac{A}{1-c} \left[ \cos\left(\phi_k - \Theta\right) - c \right]_+ \tag{1}$$

where $A$ is the response amplitude. The threshold $c$ ($-1 \leq c < 1$) determines the width of the tuning curves and hence their overlap; a large $c$ leads to narrow tuning. We assume Gaussian additive, uncorrelated noise, so that on a given trial the response of neuron $k$ is $r_k = f_k(\Theta) + \sigma\nu$, where $\sigma$ is the standard deviation of the noise, and $\nu$ is a sample from a Gaussian distribution with unit variance. We have also examined Poisson noise, and found it made no qualitative difference to our findings. We also found that a constant background added to the tuning curves did not change the bias.

We set amplitude $A = 1$, and unless denoted otherwise use $\sigma = 0.1$ and $c = -0.1$ (the cricket tuning curves in [13] were fitted with $c = -0.14$). The tuning curves are illustrated in Fig 1A. For this value of $c$, 2 neurons are simultaneously active, unless the stimulus is very close to a preferred direction of a neuron, in which case 3 neurons are active.

## Decoding methods

The task of a decoder is to estimate the encoded stimulus from the noisy response vector $\boldsymbol{r}$. We compare a number of commonly used decoders [14]:

1. The *population vector* decoder: On a given trial one first constructs the population vector

$$\boldsymbol{p} = \sum_{k=1..N} \begin{pmatrix} r_k \cos\phi_k \\ r_k \sin\phi_k \end{pmatrix} \qquad (2)$$

   From $\boldsymbol{p}$ the angle is estimated as $\hat{\theta}_{PV} = \arctan \boldsymbol{p}$. This decoder is identical to the maximum likelihood decoder in the case of dense von Mises tuning curves (see below) with Poisson noise [15,16], but in general its trial-to-trial variance is larger than for the other estimators, that is, it is not efficient (see below for the formal definition of efficiency).

2. The *maximum likelihood decoder*: According to Bayes' theorem, the posterior probability for a certain stimulus given the response is $P(\theta|\boldsymbol{r}) = P(\boldsymbol{r}|\theta)P(\theta)/P(\boldsymbol{r})$. The posterior varies from trial to trial (see Fig 1B left for some examples). Under the assumption of a uniform flat prior $P(\theta)$ and Gaussian noise, one has $\log P(\theta|\boldsymbol{r}) \propto \log P(\boldsymbol{r}|\theta) = \frac{1}{2\sigma^2}\sum_{k=1}^{N} [f_k(\theta) - r_k]^2$. The maximum likelihood estimator picks the stimulus angle that maximizes the likelihood, i.e. $\hat{\theta}_{ML} = \arg\max_\theta \log P(\theta|\boldsymbol{r})$.
   Numerically, the estimate can be found be maximizing the likelihood, but in practice it is often easier to find the maximum in a finely spaced array of candidate stimuli.

3. The *Bayesian (least squares) decoder* also relies on $P(\theta|\boldsymbol{r})$, but finds the estimate that minimizes a cost function. For the mean squared error cost, the mean of the posterior distribution minimizes the cost, hence $\hat{\theta}_{BA} = \int P(\theta|\boldsymbol{r})\theta \, d\theta$.

In the limit where many, low noise neurons are active, the likelihood becomes a Gaussian, and as a result the maximum likelihood estimator equals the Bayes estimator when the prior is uniform. However, when there are just a few noisy neurons active this is no longer true. For instance, Fig 3.7 in the textbook by Dayan and Abbott [14] shows a subtle difference in the variance of the ML and Bayesian decoders. This can only happen if the mode and mean of the posterior differ, thus hinting at a non-Gaussian, asymmetric posterior.

## Emergence of bias

To examine the emergence of bias, we sample many noisy population responses and estimate the stimulus on each trial according to these three estimators. Since the encoding model uses

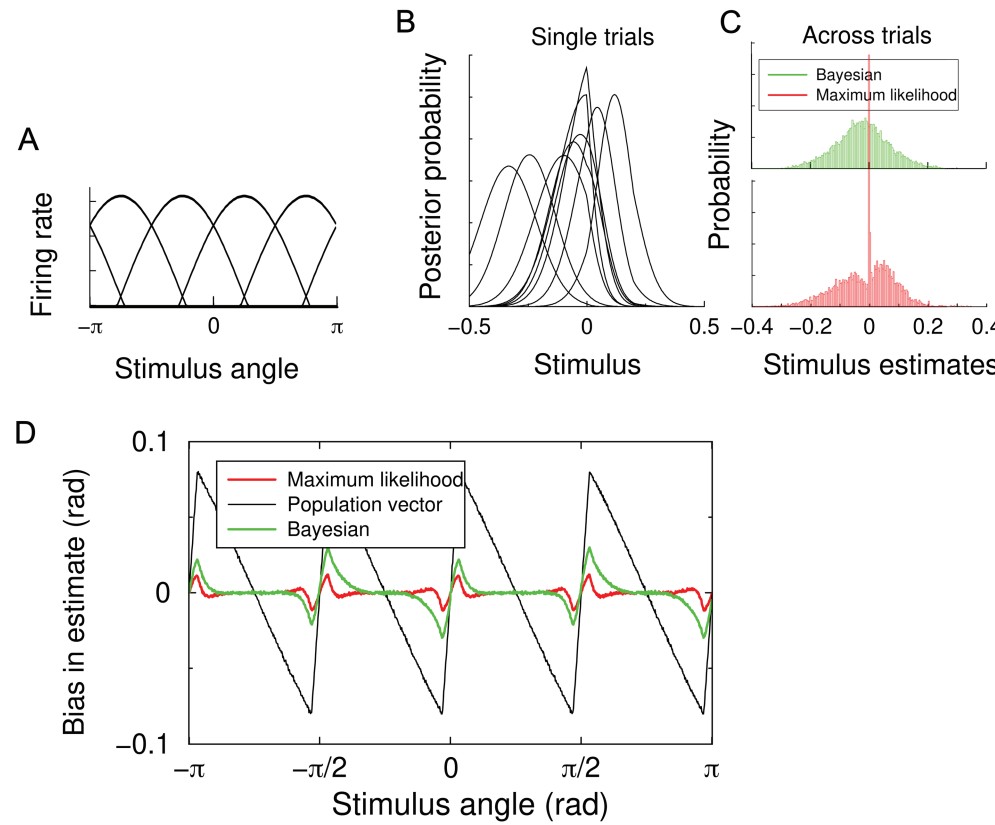

**Fig 1. Estimator bias in a model of the four neuron cricket wind direction system.** (**A**) The tuning curves of the 4 neurons as a function of the encoded stimulus angle. (**B**) Left: The posterior distributions for a few trials. On a given trial, a Bayesian estimate extracts the mean of the distribution; the maximum likelihood uses the maximum. (**C**) The distribution of estimates on a large number of trials; the bias was in this case –0.023 for the Bayesian and –0.012 for the ML decoder. The distributions are plotted relative to the true stimulus angle, which was –0.1. (**D**) Estimator bias emerges in all commonly used decoders The bias is in this case largely repulsive, i.e. away from the preferred stimuli of the four neurons (indicated by the tick marks on the x-axis).

circular functions (Eq 1), we first checked whether one needs to use circular statistics to analyze the statistics of the estimates. However, the posterior distributions are narrowly centred around true stimulus value (Fig 1C). Therefore, as long as stimuli are far from the circularity discontinuity (at 0 and $2\pi$), standard statistics and circular statistics gave identical results. Hence we only present stimuli far from the circularity discontinuities and use standard statistics throughout. Angles and biases are reported in radians.

We first fix the stimulus angle and plot the posterior distributions for a few noisy response samples in Fig 1B. The Bayesian (ML) estimator uses the mean (maximum) of the posterior to estimate the stimulus. The distribution of these estimates across many trials is shown in Fig 1C. Both estimators are biased. Interestingly, while both the ML and Bayesian estimator are biased, the ML estimator has a strong peak at the true value, absent from the Bayesian decoder. This peak is also present using our analytical method [9], and likely reflects that the maximum of the posterior is relatively stable compared to the mean.

Next, we repeat this for a number of different stimulus angles. The bias as a function of the true stimulus angle is shown in Fig 1D. Due to symmetry, the bias is an odd, periodic function, zero at the preferred angles and halfway between two preferred angles. All estimators

display a bias that happens in this case to be mostly repulsive from the preferred direction: angles slightly less than the preferred angle, are estimated as being even less. The population vector has the largest bias, while the ML and Bayesian decoders have similar size and shaped bias.

### Role of tuning curve shape and width

The tuning curve width is set by the threshold parameter $c$ in Eq 1. With wide tuning curves (thin curves in Fig 2), the bias is repulsive for all three estimators. Interestingly, as the tuning curves narrow, the bias becomes bi-phasic – some stimuli will lead to an attractive bias, others to a repulsive one. Finally for narrow tuning it is fully attractive, that is, stimuli near a preferred direction are estimated nearer to that direction.

In general, the sign of the bias is hard to intuit, but the regime of narrow tuning ($c > 0$), where stimuli close to the preferred angles activate just one neuron, is easily understood. Consider a neuron with preferred angle zero. When just this neuron is active, any off-peak response is ambiguous – the stimulus could have been on either side of the preferred stimulus: both $\Theta$, but also $-\Theta$ are equally likely. Even tiny noise will break this symmetry, hence estimating either stimulus equally. In other words, for *any* reasonable estimator the estimate will in such a case average to zero. That is, the bias is complete and attractive, $b(\Theta) = -\Theta$. Hence

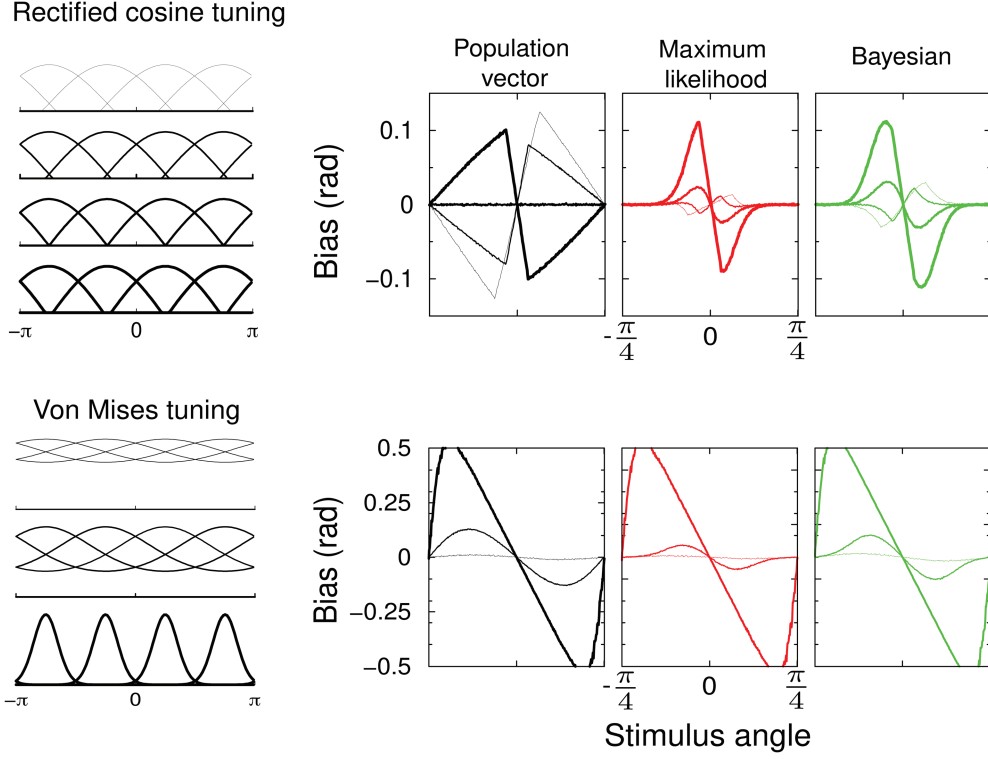

**Fig 2. Dependence of bias on tuning curve width.** Top: Estimator bias for population vector (left), maximum likelihood (middle) and Bayesian (right) estimators for 4 different tuning curve widths (illustrated on the left). Rectified cosine tuning. Thin to thick curve, $c$ = –0.2, –0.1, 0, and 0.1. Note that the bias flips sign and can be bi-phasic. Bottom: Estimator bias for von Mises tuning curves. In this case, bias is always attractive. Widths $w$ = 2 (wide; thin curve), 0.5, and 0.1 (narrow; thickest curve).

for narrow tuning the slope of the bias for all estimators is exactly –1 for angles close to zero (thick curves).

We also examine von Mises tuning curves, $f_{k=1...4}(\Theta) \propto \exp\left[(\cos(\phi_k - \Theta) - 1)/w\right]$. In this case the bias is always attractive and increases for narrow tuning widths $w$, Fig 2, bottom. These results demonstrate that (1) biases readily emerge in population codes with just a few active neurons, (2) while the three decoders always show similar biases, the dependence of the bias on tuning curve shape is intricate.

### Role of noise

In the above simulations the neural responses were noisy. Next, we investigate whether bias only occurs in the presence of neural noise. We show results for the Bayesian decoder only, as the ML decoder is similar, while the PV decoder is known to be not efficient. To summarize the complicated bias curves, we extract the minimum and maximum bias of the Bayesian decoder across the range of stimulus angles on the left of the preferred stimulus ($-\pi/4 \le \Theta \le 0$), so that a positive bias corresponds to attraction to the preferred stimulus. The minimum and maximum bias are plotted against the threshold parameter that sets the tuning width and for a number of noise levels, Fig 3A. The noise level is indicated by the line thickness (thicker

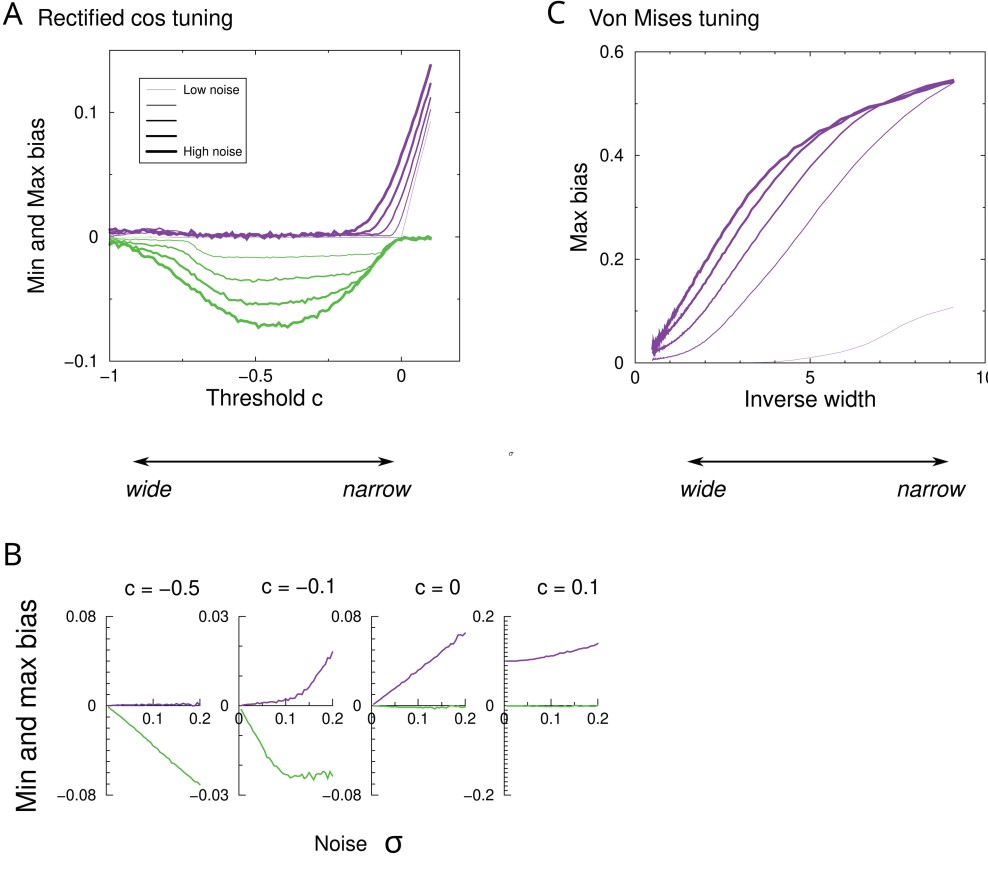

**Fig 3. Dependence of the bias on noise.** (**A**) Minimum (green) and maximum (purple) bias for the Bayesian decoder vs tuning width for 5 different levels of Gaussian noise ($\sigma$ = 0.01, 0.05, 0.1, 0.15, 0.2). Positive bias corresponds to attractive bias. (**B**) Minimum and maximum bias as a function of noise level, for different values of the threshold $c$. (**C**) As panel A, but for von Mises tuning as a function of inverse width $1/w$. In this case the bias is always positive.

lines signify more noise). At the most negative threshold ($c = -1$), the tuning curves are pure cosines (broad tuning) and bias is minimal. For intermediate thresholds ($c \approx 0.5$), the bias negative and at a given threshold, the curves in the figure are spaced out equally in the vertical direction. In other words in this regime the bias is approximately proportional to the noise. In the region $-0.2 \lesssim c \lesssim 0$, both minimum and maximum bias are non-zero; here the bias profile is bi-phasic, as can be observed in Fig 1. For positive threshold (narrow tuning) we enter the ambiguous regime described above and bias persists even in the absence of noise. The noise dependence is illustrated in detail Fig 3B.

For von Mises tuning, the bias is positive across all noise levels, Fig 3C. Again, while bias increases monotonically with noise, the dependence is non-linear (the curves are not equidistantly spaced vertically).

## Scaling with population size

The above results raise the question whether biases persist if the population contains more neurons. First we increased the density of neurons (decreasing the spacing), without changing the width or amplitude of the tuning curves. Hence the overlap increases and the decoder has access to more active neurons. This lead to a reduction in the bias, Fig 4 (black curve). Next, we scaled the tuning such that the mean activity remained constant. First, we decreased the amplitude as the number of neurons increased. When the amplitude was scaled, bias still reduced when using more neurons but less so (blue curve). As an alternative we scaled the tuning curve width, so that the average number of active neurons remains the same (red curve). In this case the bias dropped rapidly as well.

One can extend this analysis to large linear arrays. One can for instance image a linear retina where the inter-neuron distance $d$ corresponds directly to the spacing of preferred stimuli, Fig 4B. The width of the cosine tuning curves (defined as $f > 0$) equals $2d \cos^{-1}(c)$. The bias is proportional to $d$, Fig 4B. When the width is less than 2, only a single neuron is active for some stimuli; when it is less than 1, there will be stimuli that do not activate any neurons

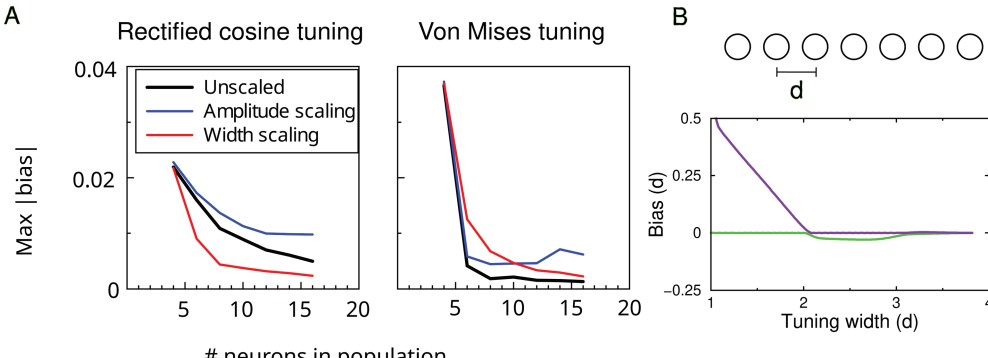

**Fig 4. Biases in larger populations.** (**A**) Maximum absolute bias vs the number of neurons in the population for the Bayesian decoder. Bias decreases with increasing neurons in the population. In the unscaled case, only the density of the neurons is increased (black curve). With amplitude scaling the mean population activity was kept the same by scaling down the response amplitude for larger population (blue curve). With width scaling this was achieving by narrowing the tuning curves. Threshold parameter $c = -0.1$ for the rectified cosine tuning with 4 neurons, and width $w$ was 1 for von Mises tuning. (**B**) Minimum and maximum bias in a long 1D array with rectified cosine tuning. Bias expressed in terms of the distance between neurons in a linear population of equally spaced neurons. The bias was calculated using the theoretical approximation.

which makes the bias ill-defined. The minimum and maximum bias curves resemble that of the 4 neuron system.

For von Mises tuning, Fig 4A right, there is a much steeper decrease for all scaling methods. While it would be interesting to examine the scaling of the bias at larger numbers of neurons, this is numerically challenging. The reason is that as the bias diminishes, it becomes comparable to the trial-to-trial fluctuations in the estimates and an unworkable large number of simulated trials would be needed.

Generally, the bias thus decreases as the number of neurons increases (with or without tuning curve normalization), but does not fully disappear.

## Decoding efficiency

Finally, we study the interaction between the bias and decoder variance. The decoder variance has been studied extensively and for finite neural populations has been analyzed in detail in [17]. The variance $\sigma_{\hat{\theta}}^2(\Theta)$ that any decoder can achieve is lower limited by the Cramer-Rao bound. For unbiased estimators, $\sigma_{\hat{\theta}}^2(\Theta) \geq /I_F(\Theta)$, where for Gaussian noise the Fisher Information is given by $I_F(\Theta) = 1/\sigma^2 \sum_k [f_k'(\Theta)]^2$. It is known that only for some estimation problems unbiased, minimal variance estimators exist [6]. Indeed, when the noise is large, the variance is larger than the bound [18]. Nevertheless in the limit of many neurons and low noise, well known decoders such as maximum likelihood decoders reach the lower bound.

When the decoder has a bias $b(\Theta)$, the bound is [6]

$$\sigma_{\hat{\theta}}^2(\Theta) \geq [1 + b'(\Theta)]^2 / I_F(\Theta) \tag{3}$$

where $b'(\Theta)$ is the derivative of the bias w.r.t. the stimulus. One defines efficiency of the estimator as the ratio of right and left hand sides of Eq 3. When it is one, the estimator reaches the bound.

As the number of active neurons is limited, the system is not homogeneous and the variance of the decoder varies with stimulus angle, Fig 5 (green curve). It drops near the preferred stimulus, as there three instead of two neurons are simultaneously active, making a more accurate estimate possible. But it increases again near zero as there the most active neuron has zero slope and would not contribute to the estimate. The bias corrected Cramer-Rao bound drops as well, so that the Cramer-Rao bound holds (green curve lies above black curve). In fact, while the efficiency is close to one away from the preferred stimulus (green and black curves are close), it drops below one near the preferred stimulus.

The inverse Fisher information also decreases near the preferred angle (again because the number of active neurons increases). It is essential to incorporate the bias correction in the bound Eq 3, as the variance of the estimator can drop below the uncorrected bound (red curve). Hence, knowing the bias is also important when evaluating the variance in the decoder.

## Approximation for Bayesian estimator bias and variance

Calculating the bias is straightforward, but compute intensive, as it requires many noisy response samples (typically 10000 in the figures and even more for larger $N$). For the maximum likelihood estimator we recently introduced a numerically exact way to calculate estimator bias and variance in the case of Gaussian noise, which gives highly precise predictions for decoding distributions and biases. Briefly, one finely discretizes the possible ML candidates and then calculates the probability that a given candidate estimate has the actual maximum likelihood from a high dimensional orthant integral, see [9] for details and software.

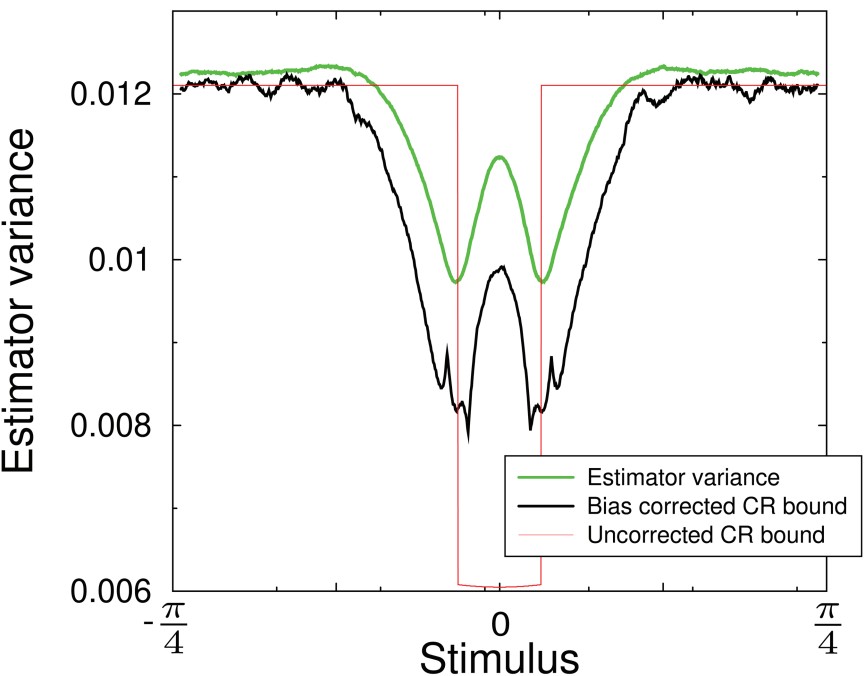

**Fig 5. Variance in the Bayesian estimator (green).** It is always larger than the Cramer-Rao lower bound (black). The inverse Fisher information $1/I_F(\Theta)$, which gives the lower bound for an unbiased estimator is also shown (red).

For the Bayesian estimator a decent approximation of the bias can be found as follows. The mean estimate is by definition

$$\left\langle \hat{\theta}_{BA} \right\rangle = \int d\boldsymbol{r}\, P(\boldsymbol{r}|\Theta) \left[ \int d\theta\, P(\theta|\boldsymbol{r})\theta \right]$$

$$= \int d\theta\, \theta P(\theta) \left[ \int d\boldsymbol{r} \frac{P(\boldsymbol{r}|\theta)P(\boldsymbol{r}|\Theta)}{P(\boldsymbol{r})} \right]$$

The normalization factor $1/P(\boldsymbol{r})$ prevents doing the integral, but we can approximate the integral by ignoring it. After completing the square, one has $P(\hat{\theta}) = \int d\boldsymbol{r} P(\boldsymbol{r}|\theta)P(\boldsymbol{r}|\Theta) \propto \exp\left[ -\sum_k \left( f_k(\theta) - f_k(\Theta) \right)^2 / 4\sigma^2 \right]$. Assuming a flat prior, the trial averaged Bayes estimator becomes

$$\left\langle \hat{\theta}_{BA} \right\rangle (\Theta) = \frac{1}{Z_1(\Theta)} \int d\theta\, \theta \exp\left\{ -\frac{1}{4\sigma^2} \sum_{k=1}^{N} \left[ f_k(\theta) - f_k(\Theta) \right]^2 \right\} \tag{4}$$

where the normalization $Z_1(\Theta) = \int d\hat{\theta}\, P(\hat{\theta})$. Circular discontinuities can be avoided by using $\hat{\theta} = \arctan\left[ \int d\hat{\theta}'\cos(\hat{\theta}')P(\hat{\theta}') / \int d\hat{\theta}'\sin(\hat{\theta}')P(\hat{\theta}') \right]$, cf. Eq 2.

Similarly, the variance in the estimator is very well approximated with

$$\sigma_{\hat{\theta}}^2(\Theta) = \frac{1}{Z_2(\Theta)} \int d\theta_1 d\theta_2\, \left( \theta_1 - \left\langle \hat{\theta}_{BA} \right\rangle \right)\left( \theta_2 - \left\langle \hat{\theta}_{BA} \right\rangle \right) Q(\theta_1, \theta_2)$$

with

$$Q(\theta_1, \theta_2) = \exp\left\{ -\frac{1}{6\sigma^2} \sum_k \left[ f_k(\theta_1) - f_k(\Theta) \right]^2 + \left[ f_k(\theta_2) - f_k(\Theta) \right]^2 + \left[ f_k(\theta_1) - f_k(\theta_2) \right]^2 \right\}$$

and normalization $Z_2(\Theta) = \int d\theta_1 d\theta_2 \, Q(\theta_1, \theta_2)$.

These calculations are easily extended to the case of Poisson noise. There one finds

$$\langle \hat{\theta}_{BA} \rangle (\Theta) = \frac{1}{Z_1(\Theta)} \int d\theta \, \theta \prod_k I_0 \left( 2T \sqrt{f_k(\theta) f_k(\Theta)} \right)$$

where $I_0$ is the modified Bessel function of the first kind, $f$ is again the tuning curve but now expressed as a firing rate, and $T$ is the duration of the spike count window. Likewise

$$\sigma_{\hat{\theta}}^2(\Theta) = \frac{1}{Z_2(\Theta)} \int d\theta_1 d\theta_2 \prod_k {}_0F_2 \left( ; 1, 1; T^3 f_k(\theta_1) f_k(\theta_2) f_k(\Theta) \right) \exp \left[ -T(f_k(\theta_1) + f_k(\theta_2) + f_k(\Theta)) \right]$$

where the generalized hypergeometric function ${}_0F_2(; 1, 1; x) = \sum_k x^k / (k!)^3$.

To examine the accuracy of this approximation we compare it to the simulation results above, repeating the parameter variations, Fig 6. The approximation (dashed curves) tends

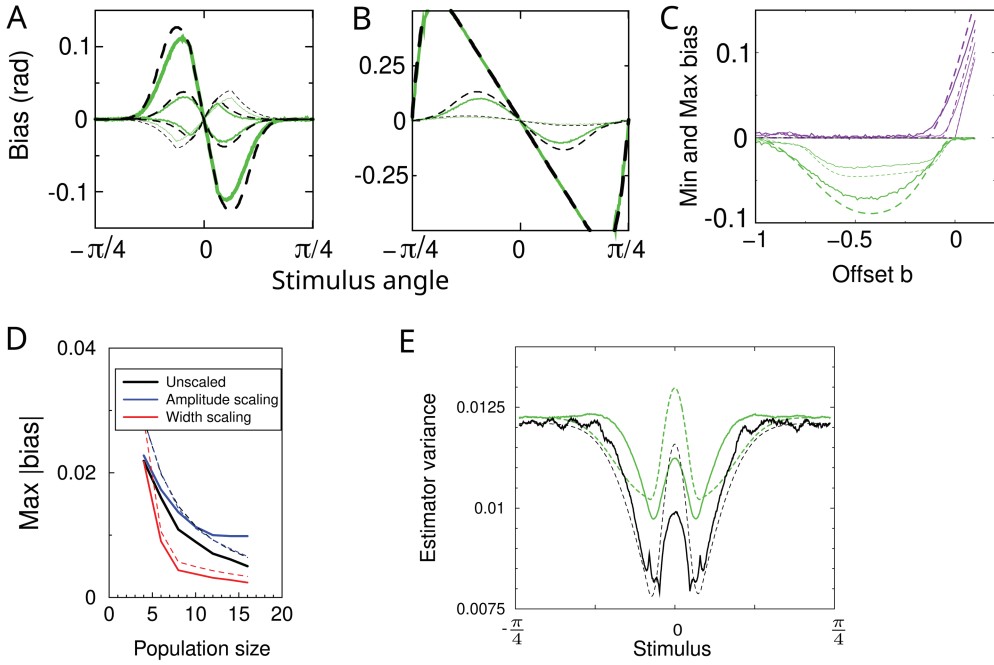

**Fig 6. Analytical approximation of the bias and variance of the Bayesian decoder.** Simulation results, replotted from above figures, are shown as solid lines, the approximations are shown as dashed lines. Parameters and plotting conventions as in previous figures. (A) The bias for rectified cos tuning for a number of different offset parameters ($c = -0.2, -0.1, 0, 0.1$). (B) The bias for von Mises tuning ($w = 2, 0.5, 0.1$). (C) Minimum and maximum bias of cosine rectified tuning for different noise levels (3 different levels of Gaussian noise, $\sigma = 0.01, 0.1, 0.2$; two intermediate noise levels were omitted for clarity). (E) Bias against number of neurons across the three scaling methods. (E) Estimator variance and Cramer-Rao bound.

to moderately overestimate the bias, displaying a larger magnitude than the simulation (solid curves). However, the approximation follows qualitatively all the features observed in the simulation, including its noise dependence and bi-phasic bias.

## Discussion

In summary, population codes that activate just a few neurons are prone to decoding biases. The bias is relevant for systems with just a few neurons i.e. in simple nervous systems found in insects, but also in large neural arrays with narrow tuning so that only a few neurons are activated (as can occur in the retina). In particular, the biases of the ML decoder and the Bayesian decoder are similar in size and parameter dependence. But also the population vector shows bias, thus the emergence of bias appears general. Surprisingly, while for von Mises tuning curves the bias is always attractive, for rectified cosine tuning curves the sign of the bias can be either attractive, repulsive, or even bi-phasic. Moreover, while noise always increases bias, the dependence is non-linear and for narrow tuning bias can persist at zero noise.

One can wonder if the biases could be observable experimentally. There are two challenges, First, the bias is at most some 30% of the standard deviation, see for example the histograms in Fig 1C. However, in contrast to the bias, the variance will average out. This means that experiments would need a sufficiently large number of trials to estimate the bias. Second, the magnitude of the bias is smaller than the difference in preferred stimuli of neighboring neurons, Fig 4B. Yet, in the field of visual hyperacuity (perception with a scale finer than the classical resolution), it is common to observe effects 10 times smaller than the neural stimulus spacing [19], hence experimental confirmation might be possible.

It is unfortunately not easy to gain intuition in the cause and sign of bias (with the exception of the limit of narrow tuning). The complicated dependence of the bias on tuning curve properties and noise hinder analytical treatment in even simple cases. In addition, the decoders require many trials to get a good idea of their average behavior, and can have problems such as local minima in a complex loss landscape. To partly mitigate this issue, we have introduced an approximation to the bias and variance in the Bayesian decoder. While an approximation that moderately overestimates the bias, it captures all observed dependencies. The approximation is efficient as it is just a single integral (per stimulus) and can be carried out with standard integration routines. Also for the ML decoder the bias can be calculated without relying on simulation [9]. In contrast, that method is very accurate but still quite computationally demanding as it relies on Monte Carlo integration. The approximation introduced here allows for a rapid calculation of biases, sufficiently accurate for exploratory purposes. While neither analytical approach necessarily allows for more analytical understanding of e.g. the link between tuning properties and bias, they provide a further evidence that the biases we report are fundamental properties of the systems we studied.

The bias is largest for narrow tuning curves. In contrast, it is well known that decoder accuracy increases with narrow tuning [20,21]. The reason is that steep tuning curves yield high sensitivity to small stimulus changes, which reduces variance in the decoder. Thus achieving small decoding bias and small decoding variance might be biologically competing objectives.

Because the bias is a deterministic function, it is in principle possible to create a decoder that inverts the bias and thus compensates for it. When the bias is constant, compensation is trivial. But here the bias varies with the true stimulus and moreover, the compensation needs to be aware of noise, i.e. the observation time. If a mismatched bias correction is used, the cure might be worse than the disease. In [9] a Tikhonov regularized bias compensation was employed, which indeed reduced bias, however it led to a large increase in the variance of the decoder.

We note that in the nervous system, subsequent processing stages do not *read out* the code, but instead *process* the population code. It would therefore be interesting to consider in the future how biases propagate through networks. Finally we note that biases might also have functional benefits in particular when they are noise dependent. When it is crucial to never underestimate a sensory stimulus, say, in collision avoidance, a noise-dependent bias might lead to an adaptive safety margin. It will be interesting to examine whether such effects are exploited in biology.

## Materials and methods

All simulations and decoding methods were constructed in Octave [22]. Code is available at https://github.com/vrossumlab/bias24.

## Acknowledgments

It is a pleasure to thank Nikos Gekas for discussion.

## Author contributions

**Conceptualization:** Sander W Keemink.

**Formal analysis:** Mark C. W. van Rossum.

**Investigation:** Sander W Keemink, Mark C. W. van Rossum.

**Software:** Sander W Keemink.

**Visualization:** Sander W Keemink, Mark C. W. van Rossum.

**Writing – original draft:** Sander W Keemink, Mark C. W. van Rossum.

**Writing – review & editing:** Sander W Keemink, Mark C. W. van Rossum.

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
