## [Decision Letter · Decision Letter 0]

25 Nov 2024

PCOMPBIOL-D-24-01599Biases in population codes with a few active neuronsPLOS Computational BiologyDear Dr. van Rossum,

Thank you for submitting your manuscript to PLOS Computational Biology. After careful consideration, we feel that it has merit but does not fully meet PLOS Computational Biology's publication criteria as it currently stands. Therefore, we invite you to submit a revised version of the manuscript that addresses the points raised during the review process.

Please submit your revised manuscript within 60 days Jan 25 2025 11:59PM. If you will need more time than this to complete your revisions, please reply to this message or contact the journal office at ploscompbiol@plos.org. Please include the following items when submitting your revised manuscript: * A rebuttal letter that responds to each point raised by the editor and reviewer(s). You should upload this letter as a separate file labeled 'Response to Reviewers'. This file does not need to include responses to formatting updates and technical items listed in the 'Journal Requirements' section below. * A marked-up copy of your manuscript that highlights changes made to the original version. You should upload this as a separate file labeled 'Revised Manuscript with Track Changes'. * An unmarked version of your revised paper without tracked changes. You should upload this as a separate file labeled 'Manuscript'.

We look forward to receiving your revised manuscript. Kind regards, Stefano PanzeriAcademic EditorPLOS Computational Biology  Hugues BerrySection EditorPLOS Computational BiologyFeilim Mac GabhannEditor-in-ChiefPLOS Computational Biology

Jason Papin

Editor-in-Chief

PLOS Computational Biology

**Journal Requirements:**

At this stage, the following Authors/Authors require contributions: Sander Keeming, and Mark C. W. van Rossum. Please ensure that the full contributions of each author are acknowledged in the "Add/Edit/Remove Authors" section of our submission form.

3) Please provide an Author Summary. This should appear in your manuscript between the Abstract (if applicable) and the Introduction, and should be 150u2013200 words long. The aim should be to make your findings accessible to a wide audience that includes both scientists and non-scientists. Sample summaries can be found on our website under Submission Guidelines:

4) Your manuscript is missing the following sections: Introduction, Results, and Methods.  Please ensure all required sections are present and in the correct order. Make sure section heading levels are clearly indicated in the manuscript text, and limit sub-sections to 3 heading levels. An outline of the required sections can be consulted in our submission guidelines here:

5) Please upload all main figures as separate Figure files in .tif or .eps format. For more information about how to convert and format your figure files please see our guidelines:

**Reviewers' comments:**

Reviewer's Responses to Questions

**Comments to the Authors:**

Reviewer #1: The manuscript makes the point that decoding the value of a correlate of neural activity from only a few neurons in a population implies, alongside random variability, a systematic bias, which should be taken into account when extrapolating experimental measures to the conceptual limit of the code expressed by a vary large population of neurons.

This is certainly a valid point. It does not seem to me particularly original or innovative (any non-linear function of probabilities estimated through frequencies is bound to be affected by some systematic bias), but I would not be able to cite a paper where it has been made with compelling clarity. Therefore, I am in principle sympathetic to this study.

The study does not, however, deliver on the expectations it has raised. It examines a rather specific instance of the cosine coding of an angle – the external correlate – by neurons that have each their own preferred angle. The rather artificial case is considered of 4 clusters of neurons having their preferred angles S, E, N, W, which had been introduced in the literature, indeed, but not everything that has been published before need be particularly profound. A number of decoding schemes are examined, including “population vector”, “maximum likelihood” and “Bayesian” estimators of the true angle. The manuscript then proceeds by showing us an abundance of figures generated by computer simulations of the combined encoding-decoding system, which show a variety of behaviors of the resulting systematic bias. There is no attempt to analyze things mathematically, which may in fact not be feasible. There are no general messages either, at least none that I was able to evince myself looking at the multiple figures. Therefore, I would regard the study as a valuable control exercise, that one would need to do when applying decoding procedures to neural data and before drawing hasty conclusions; but not as a standalone scientific contribution that deserves publication in itself, given that the exercise would have to be repeated for the different instances of encoding and decoding one would want to consider.

Reviewer #2: Summary of the content:

This paper analyzes the bias that arises during the decoding process in population coding when the number of neurons is small. The analysis focuses on the following aspects: (1) comparison of two different tuning curves—rectified cosine tuning and Von Mises tuning—and their impact on bias; (2) the effect of tuning curve width on bias; (3) the influence of noise intensity on bias; and (4) the impact of the number of neurons. Finally, the paper proposes an approximate method for calculating bias and variance, comparing it with simulation results.

Strengths:

(1) Previous studies on population coding typically assume a sufficient number of neurons, allowing the decoding process to be considered unbiased. This paper highlights the significance of studying population coding with a small number of neurons, which is relevant to certain sensory systems in insects. This offers an interesting perspective on the topic.

(2) The paper presents an intriguing observation: unlike Von Mises tuning, rectified cosine tuning exhibits three types of bias tendencies—repulsive, attractive, and bi-phasic. While the attractive bias is intuitive, the appearance of the other two tendencies is unexpected, adding depth to the discussion.

Weaknesses:

(1) The theoretical analysis does not provide deeper insights into the simulation results. The approximate method proposed at the end of the paper, regardless of its validity, still leads to expressions containing integrals. These do not directly reflect the influence of key factors discussed earlier, such as tuning curve shape, tuning width, noise level, and neuron number, on the bias.

(2) The paper lacks in-depth analysis of the simulation results. While the simulations are reported, there is insufficient exploration of the underlying mechanisms. For instance, the paper mentions that Von Mises tuning only results in attractive bias, whereas rectified cosine tuning exhibits three bias tendencies (repulsive, attractive, and bi-phasic). However, the authors only analyze the cause of the attractive bias, leaving the other two phenomena unexplained.

(3) Some of the paper’s descriptions are imprecise. For example, on page 6, it states: "For intermediate thresholds (c ≈ 0.5), the bias is negative, and at a given threshold, the curves in the figure are spaced out equally in the vertical direction, in other words, the bias is proportional to the noise." I disagree with the claim that the curves are equally spaced. The authors neither provide simulation evidence of the linear relationship between bias and noise nor offer a theoretical justification. This statement feels more like an unsubstantiated assumption.

(4) The figures are poorly constructed. Most of the figures fail to clearly convey their intended meaning. For instance, in Figure 6, all approximate theoretical results are represented by black dashed lines without distinctions in thickness or color, making it impossible for readers to compare theoretical and simulation results.

Conclusion:

While the paper's premise and some of its simulation findings are novel, it falls short of being a polished scientific paper. At best, it reads as a report summarizing a set of simulation results. The authors need to conduct a more thorough analysis of the observed phenomena and significantly improve the clarity of their figures and descriptions to enhance the paper’s readability and scientific rigor.

Reviewer #3: This paper describes an analysis of population coding and decoding in small neural populations in which only a few neurons are active in response to a given stimulus. Most of it focuses on a 4-neuron setup inspired by the cricket wind sensor system. The paper focuses primarily on bias of estimators decoding this population code, considering three different estimators (population vector, maximum likelihood, and Bayes' least squares) across a variety of tuning curve shapes, widths, and noise levels. The authors briefly consider decoder variance and a comparison to Fisher Information and the Cramer Rao bound. In the final section, they derive an analytic approximation to the bias and variance of the Bayesian decoder.

Overall, the paper makes a worthwhile contribution to the literature, although in many cases it appears the bias is so tiny for both ML and Bayesian estimators (eg Figs 1-4) that I found myself wondering if this level of bias is something that is worth caring about. It would be helpful if the authors could add some comments about what they consider to be a realistic setting (eg for tuning curve width and noise level), and whether they expect the bias they observe to have any meaningful behavioral consequences.

I found two significant shortcomings that I think should be addressed before the paper is suitable for publication.

1) Circular variables. The paper repeatedly invokes the Gaussian distribution, but it focuses primarily on a setting with circular variables, where the (standard) Gaussian distribution cannot be applied. It may be the case that the posterior distribution is narrow and can be well approximated locally by a Gaussian in many settings, but it is still not technically Gaussian. So the authors should take more care with the appropriate treatment of circular variables. Similarly, the mean is not well defined for circular variables. (The circular mean can be used to obtain an estimate of an angular variable, but it is not the same as the standard mean -- I left a few more detailed remarks in the comments below). Overall, the paper could benefit from greater rigor on this point.

2) Variance. The paper focuses primarily on estimator bias; it considers variance only in Figure 5, using a single choice of tuning curve and noise level. However, I found myself thinking that in the case of the narrow tuning curves shown in Fig 2, the effects on variance might be far larger than those on bias. (It would also be nice to see if the Bayesian estimator achieves lower variance than the ML estimator, since we have already seen that in some cases it exhibits higher bias). Overall, it seems the paper would be more substantial if the paper gave a more equal treatment of estimator variance (or even MSE), which I suspect might be more interesting than bias in some cases.

Detailed comments:

-----

Abstract: "codes can also display biases"

- this strikes me as non-standard usage of the word "bias". The technical definition of bias applies to estimators not "codes". (It's not clear to me what it would mean for a code to be biased; rather the estimators based on this code could be biased or unbiased.)

-----

pg 2: "minimizing the likelihood" -> "maximizing the likelihood".

-----

pg 2: "this is the mean of the distribution". -> "... posterior distribution".

-----

pg 2: "The Bayesian decoder"

This estimate (also known as "Bayes least squares") is perfectly fine in general terms, but it's worth noting that this formula is not applicable as written to circular variables like orientation.

Unfortunately, the standard arithmetic mean isn't applicable to circular data. You could take the circular mean, which can be defined using complex numbers, although in this case the distance being minimized is slightly different than the Euclidean mean. See eg: https://en.wikipedia.org/wiki/Circular_mean.

-----

Fig 1A: This plot is relatively difficult to grok, and obscures the amplitude of the tuning curves. I would prefer to see the tuning curves plotted with "expected response" on the y axis and orientation theta on the x axis. The current plot lacks axis labels, so it's unclear what the 5 circles correspond to. (It would be helpful to label the tuning curves for different neurons, eg "neuron 1", "neuron 2", etc).

-----

Fig 1B. The caption says "Left: The estimate distribution for a few trials". But these look like posterior distributions (or perhaps likelihoods), not estimates. (I would think "estimate distribution" refers to a distribution of estimates, which is shown on the right). he estimate for a few trials would just be a few points.

If these are likelihoods or posteriors, the x axis label should be "stimulus" not "stimulus estimate". (In this case the peak of each curve would be the ML or MAP estimate).

Right side of Fig 1B, why is there a big spike in the histogram for the ML estimate? It's not obvious to me that this should occur -- is it possible there's an error due to finite gridding of the likelihood?

-----

Fig 1B caption: "A Bayesian estimate extracts the mean of the distribution".

Once again, this is not quite correct for circular variables. (The cost function minimized by the circular mean is not the standard arithmetic mean, as noted above).

-----

Fig 1B caption: "the arrow indicates the average estimate, i.e. the bias."

Sorry, something is wrong here. The black arrow looks nowhere close to the mean of the black histogram. How can this be the average estimate?

There's also a problem that the red arrow is not visible against the red histogram.

-----

Fig 1C: Y axis needs units. (Presumably the bias here is shown in units of radians?)

Same comment applies to other figs showing bias (eg Fig 2, Fig 3, etc).

-----

Fig 1C caption: "away from the preferred stimuli". Would be clearer to say "away from the preferred stimuli of the four neurons in the population."

-----

Pg 4, "Emergence of Bias" section. Can the authors give any insight into why the estimate is biased here? Is it because only one neuron is active at the 4 cardinal orientations? (It's a little bit hard to tell from the tuning curves in Fig 1A if this is the case or not).

Fig 2 and the next section do examine how tuning curve width affects the bias, but the bias ends up feeling mysterious here. If the tuning curves overlapped more I would expect this bias to go to zero. (What happens if the noise goes to zero -- does the bias disappear or is it still present? It seems the authors address this in Fig 3, but I didn't see them explicitly address whether bias goes to zero in the limit sigma -> 0.)

The following section gives an explanation for why there should be an attractive bias in the case of narrow tuning curves, but it still seems mysterious that there is a bias when the tuning curves are broad. Can the authors provide any intuition for why there's a bias in this case?

-----

pg. 4: "The likelihood becomes Gaussian". Again, this is not quite right for circular variables.

(This whole section needs a bit more care in the case of circular variables).

-----

pg. 4: "However, when there are just a few noisy neurons active this is no longer true. For instance, Figure 3.7 in the textbook by Dayan and Abbott (2001) shows a subtle difference in the variance of the ML and Bayesian decoders, hinting at a non-Gaussian posterior distribution. Also here the distribution of estimate is non-Gaussian, Fig.1B right."

Rather than focusing only on difference in the variance of the estimators, it might be helpful to remind readers why the ML estimate is not the same as the Bayesian estimate (even when the prior is uniform): it's because the mode of the posterior (which is also the mode of the likelihood) is not necessarily equal to the mean of the posterior. And the mean (but not the mode) of the posterior achieves minimum MSE. So in cases where the posterior is not symmetric, the mean and mode can be in different locations.

------

pg 6: "First we increased the number of neurons without changing any of the tuning curve properties, "

I'm not sure what this means. Do you mean you replicated the existing tuning curves so there are now multiple copies of each of the 4 canonical tuning curves? Or you added tuning curves with the same shape but with different preferred orientations to create a population that tiles?

-------

pg 6. "bias only depends weakly on the signal-to-noise ratio of the neurons, consistent with the above observations."

This seems to be contradicted by Fig 3, which shows (for both kinds of tuning curves) that bias grows much smaller with reduced niose. Can you clarify?

-----

pg 7: typo: "variance of the decoder varies across with the stimulus"

**Have the authors made all data and (if applicable) computational code underlying the findings in their manuscript fully available?**

Reviewer #1: **No: **I do not think the question is relevant in this case. The codes are rather trivial.

Reviewer #2: Yes

Reviewer #3: **No: **It would be nice to provide a link to a github repo with code to reproduce these experiments. (Apologies if I missed it).

PLOS authors have the option to publish the peer review history of their article (what does this mean?). If published, this will include your full peer review and any attached files.

Reviewer #1: No

Reviewer #2: No

Reviewer #3: No

**Figure resubmission:** While revising your submission, please upload your figure files to the Preflight Analysis and Conversion Engine (PACE) digital diagnostic tool, https://pacev2.apexcovantage.com/. PACE helps ensure that figures meet PLOS requirements. To use PACE, you must first register as a user. Registration is free. Then, login and navigate to the UPLOAD tab, where you will find detailed instructions on how to use the tool. If you encounter any issues or have any questions when using PACE, please email PLOS at figures@plos.org. Please note that Supporting Information files do not need this step. If there are other versions of figure files still present in your submission file inventory at resubmission, please replace them with the PACE-processed versions.
---

## [Decision Letter · Decision Letter 1]

24 Feb 2025

PCOMPBIOL-D-24-01599R1

Biases in neural population codes with a few active neurons

PLOS Computational Biology

Dear Dr. van Rossum,

Thank you for submitting your manuscript to PLOS Computational Biology. After careful consideration, we feel that it has merit but does not fully meet PLOS Computational Biology's publication criteria as it currently stands. Therefore, we invite you to submit a revised version of the manuscript that addresses the points raised during the review process.

Please submit your revised manuscript within 30 days Apr 26 2025 11:59PM. If you will need more time than this to complete your revisions, please reply to this message or contact the journal office at ploscompbiol@plos.org. Please include the following items when submitting your revised manuscript:

We look forward to receiving your revised manuscript.

Kind regards,

Stefano Panzeri

Academic Editor

PLOS Computational Biology

Hugues Berry

Section Editor

PLOS Computational Biology

**Additional Editor Comments:**

Dear Authors, thanks for the careful revision. Only small steps are needed to complete the revision of the paper. Please consider the final suggestions of the Reviewer, which require little work but which we feel will improve the manuscript. Thanks again for this nice work. Stefano

**Journal Requirements:**

1) Please provide an Author Summary. This should appear in your manuscript between the Abstract (if applicable) and the Introduction, and should be 150-200 words long. The aim should be to make your findings accessible to a wide audience that includes both scientists and non-scientists. Sample summaries can be found on our website under Submission Guidelines:

2) Your manuscript is missing the following sections: Introduction, Results, and Methods.  Please ensure all required sections are present and in the correct order. Make sure section heading levels are clearly indicated in the manuscript text, and limit sub-sections to 3 heading levels. An outline of the required sections can be consulted in our submission guidelines here:

3) Thank you for stating "Code is available at https://github.com/vanrossumlab/bias24".This link reaches a 404 error page. Please amend this to a working link or provide further details to locate the code."

4) Please upload the figures in a correct numerical order in the online submission form.

**Reviewers' comments:**

Reviewer's Responses to Questions

Reviewer #3: The authors have done a nice job addressing my comments and I thank them for their additional analyses and clarifications, which make the paper much stronger.

I have only one lingering comment, which is that I would encourage the authors to provide a bit more extensive discussion of the implications of using Gaussian distributions to describe circular variables. The section about this caveat currently reads (pg 3):

"First, we checked whether one needs to use circular statistics to analyze the statistics of the estimates. Because the posterior distributions are narrow linear and circular statistics gave identical results (as long as decoded angles stayed away from the 2π discontinuity), we will use linear statistics throughout."

The phrase "linear statistics" is rather opaque, and does not provide enough detail to describe what the authors actually did. I would like to see the authors expand this to say something like "Thus, when we refer to Gaussian distributions over orientation, we mean a Gaussian distribution over the interval [mu-pi, mu+pi], where mu is the mean of the distribution. In this setting, the probability mass on [-infinity, mu-pi), and [mu+pi, infinity] is negligible and can be safely ignored." (If that is indeed what the authors have in mind -- I'm honestly not sure what exactly the authors are doing when they refer to "linear statistics", as there are a variety of ways to formulate distributions on the unit circle (e.g., wrapped Gaussians vs. truncated Gaussians vs. von Mises distributions). But the text should have enough mathematical detail that a reader would know how to handle a case where the oriention of interest is say 2pi - 0.1. How do they go about formulating the linear decoder in this case? (This could be relegated to a Methods section, if necessary).

**Have the authors made all data and (if applicable) computational code underlying the findings in their manuscript fully available?**

Reviewer #3: Yes

PLOS authors have the option to publish the peer review history of their article (what does this mean?). If published, this will include your full peer review and any attached files.

Reviewer #3: No

**Figure resubmission:**
---

## [Editor Report · Decision Letter 2]

14 Mar 2025

Dear Dr. van Rossum,

We are pleased to inform you that your manuscript 'Biases in neural population codes with a few active neurons' has been provisionally accepted for publication in PLOS Computational Biology.

Best regards,

Stefano Panzeri

Academic Editor

PLOS Computational Biology

Hugues Berry

Section Editor

PLOS Computational Biology

---

## [Editor Report · Acceptance letter]

PCOMPBIOL-D-24-01599R2

Biases in neural population codes with a few active neurons

Dear Dr van Rossum,

I am pleased to inform you that your manuscript has been formally accepted for publication in PLOS Computational Biology. Your manuscript is now with our production department and you will be notified of the publication date in due course.

With kind regards,

Olena Szabo
